

# Prediction of coronary heart disease in rural Chinese adults: a cross sectional study

Qian Wang[1,*], Wenxing Li[2,*], Yongbin Wang[1], Huijun Li[1], Desheng Zhai[1] and Weidong Wu[1]

[1] School of Public Health, Xinxiang Medical University, Xinxiang, Henan, China
[2] Department of Biochemistry and Molecular Biology, School of Basic Medical Sciences, Southern Medical University, Guangzhou, Guangdong, China
* These authors contributed equally to this work.

## ABSTRACT

**Background:** Coronary heart disease (CHD) is a common cardiovascular disease with high morbidity and mortality in China. The CHD risk prediction model has a great value in early prevention and diagnosis.

**Methods:** In this study, CHD risk prediction models among rural residents in Xinxiang County were constructed using Random Forest (RF), Support Vector Machine (SVM), and the least absolute shrinkage and selection operator (LASSO) regression algorithms with identified 16 influencing factors.

**Results:** Results demonstrated that the CHD model using the RF classifier performed best both on the training set and test set, with the highest area under the curve (AUC = 1 and 0.9711), accuracy (one and 0.9389), sensitivity (one and 0.8725), specificity (one and 0.9771), precision (one and 0.9563), F1-score (one and 0.9125), and Matthews correlation coefficient (MCC = one and 0.8678), followed by the SVM (AUC = 0.9860 and 0.9589) and the LASSO classifier (AUC = 0.9733 and 0.9587). Besides, the RF model also had an increase in the net reclassification index (NRI) and integrated discrimination improvement (IDI) values, and achieved a greater net benefit in the decision curve analysis (DCA) compared with the SVM and LASSO models.

**Conclusion:** The CHD risk prediction model constructed by the RF algorithm in this study is conducive to the early diagnosis of CHD in rural residents of Xinxiang County, Henan Province.

## INTRODUCTION

Cardiovascular disease remains the top cause of death in China, taking an estimated 3.97 million lives in 2016 (*Liu et al., 2019*). Coronary heart disease (CHD) is to blame for a large share of the deaths, and the number of deaths is 232.2 deaths per 100,000 population. The incidence of CHD grows rapidly and will remain an upward trend in the next decade. The increasing burden of CHD has become a major public health problem in China (*Ma et al., 2020*). Although massive medical resources and technologies are used to treat this disease, recent studies have found that early prevention plays a vital role in reducing

Corresponding author
Weidong Wu, wdwu2013@126.com

the incidence and death of CHD in many developed countries (*Benjamin et al., 2019*; *Unal, Critchley & Capewell, 2004*). CHD develops slowly, usually over decades, so early prevention and diagnosis have become immensely important in improving long-term survival of patients with CHD (*Richard, 2011*).

The establishment of a risk prediction model is of great significance for the early prevention and diagnosis of CHD (*Chen et al., 2021*; *Jee et al., 2014*; *Savira et al., 2021*). The earliest well-known model of CHD was the Framingham model, which was developed for assessing the 10-year risk of developing CHD. Sex, age, blood pressure, smoking behavior, cholesterol (TC), low-density lipoprotein cholesterol (LDL_C), high-density lipoprotein cholesterol (HDL_C), and diabetes status were considered as CHD risk factors in their study (*Wilson, D'Agostino & Levy, 1998*). Since the introduction of the Framingham Risk Score (FRS), countless CHD risk predictive models have been developed. For example, the American Heart Association (ACC) and the American Heart Association (AHA) developed ACC/AHA pooled cohort equations risk calculator to assess 10-year atherosclerotic cardiovascular disease (ASCVD) risk (*Goff et al., 2013*; *Grundy et al., 2019*); In the UK, the JBS3 risk calculator was designed by the joint British societies to estimate 10-year risk of cardiovascular disease as well as lifetime risk of cardiovascular disease in all individuals (*Board, 2014*). In China, *Yang et al. (2016)* developed the China-PAR equations for risk assessment of ASCVD among northern and southern China populations. These models were mostly derived from the Framingham study and performed well when applied to populations with similar high background cardiovascular disease risk (*Al-Shamsi, 2020*; *D'Agostino et al., 2001*; *Zheng et al., 2019*). However, studies of *Bhatnagar (2017)* and *Kreatsoulas & Anand (2010)* have found that social environments, ethnic and geographic differences can cause changes in risk factors for CHD. These tools often overestimate or underestimate CHD risk in new cohorts due to regional and ethnic restrictions (*Ganz et al., 2016*; *Hermansson & Kahan, 2018*; *Santos-Ferreira et al., 2020*; *Wallisch et al., 2020*).

With the development of machine learning technology and the expansion of its application fields, machine learning for data mining in medicine and health care has become a research hotspot (*Alanazi, Abdullah & Qureshi, 2017*; *Giger, 2018*; *Vellido, 2019*). Compared with traditional statistical methods, machine learning can mine medical data with higher dimensions, complex structures, and large amounts (*Ahmed, Sankar & Sandhya, 2021*; *Yue et al., 2020*). Machine learning algorithms such as Naive Bayes (*Pouriyeh et al., 2017*), decision tree (*Tayefi et al., 2017*), neural network (*Karayılan & Kılıç, 2017*), Random Forest (RF) (*Singh, Sinha & Singh, 2017*), and Support Vector Machine (SVM) (*Gokulnath & Shantharajah, 2019*) have been proven effective in their capability of diagnosis of heart disease.

In this study, the least absolute shrinkage and selection operator (LASSO) regression and two machine learning methods (SVM and RF) were used to develop a CHD risk prediction model. Data comes from the 2017 survey of chronic non-communicable diseases in Xinxiang County, Henan Province of China. Confusion matrix, net reclassification index (NRI), integrated discrimination improvement (IDI), decision curve analysis (DCA) and receiver operating characteristic (ROC) curve were adopted to

evaluate the performance of different algorithms on predicting CHD among rural residents in Xinxiang county. Moreover, CHD influencing factors for residents in Xinxiang County were also identified.

## MATERIALS & METHODS

### Data collection

In this research, 10,691 residents from two townships (Lang Gong Temple township and seven Li Ying township) of Xinxiang county, Henan province, in central China were selected by multistage randomized cluster sampling and investigated with questionnaire survey, physical examination, and lab tests. The participants ranged in age from 18 to 79, with 4,352 males and 6,339 females. All participants received a written informed consent before participating in this study. Questionnaire surveys included demographic information (name, ID, address, contact number, gender, ethnicity, occupation, date of birth, civil state, income, and medical services, etc.), lifestyles (smoking status, alcohol intake, eating habits, tea drinking and athletic activity, etc.), sleep situation, personal and family history of 20 common chronic non-communicable diseases (hypertension, hyperlipidemia, diabetes, CHD, stroke, emphysema, gastrointestinal disease, allergic rhinitis, chronic bronchitis, asthma, kidney disease, gallbladder disease, chronic obstructive pulmonary disease, cancer, liver disease, chronic hepatitis, pancreatic disease, skin disease, gout, and tuberculosis), menstrual, reproductive history and gynecological diseases (for women) and psychologic status. Anthropometric data included height, waistline, hipline, weight, body fat rate (BFR), basal metabolic rate (BMR), visceral fat index (VFI), grip strength, and heart rate (HR). Lab tests included blood routine (24 items), blood lipids, blood glucose (fasting blood glucose (FBG), fasting insulin levels (FINS), and glycated hemoglobin A1c (HbA1c)), hepatic and kidney function, resting blood pressure, urine routine, heavy metal in urine, stool routine, bone mineral density, lung function, 12-lead resting electrocardiography, chest X-ray, and abdominal ultrasonography.

The datasets of this study were derived from a cross-sectional survey of common chronic noncommunicable diseases in rural areas of Henan, China, conducted by Xinxiang Medical University Health Team. All methods were carried out in conformity to the relevant guidelines and regulations approved by the Ethical Committee of Xinxiang Medical University for Human Studies (protocol number XYLL-2016176, approved 9 June 2016) and all participants received written informed consent. Besides, this research meets all the guidelines in the Declaration of Helsinki.

### Data preprocessing

This part was to preprocess all the data of 10,691 participants collected above. This dataset had a large number of missing and repeated values, contained variables that were not related to CHD, and the data storage format was inconsistent. Therefore, it was necessary to preprocess the original data to improve the accuracy of subsequent analysis. Process of preparing data for prediction models for CHD comprises the following: (1) Data selection. First, to develop the diagnosis model of CHD in central Chinese rural adult populations, 915 patients with CHD and 2,829 residents without 20 common chronic
non-communicable diseases were selected for analysis based on personal disease history. Next, due to the large amount of information contained in the datasets, we only extracted part of the attribute data closely related to CHD for analysis and processing. The variables included ID, gender, age, smoking status, alcohol intake, psychological status, height, hip circumference, weight, BMR, waist circumference, VFI, grip strength, BFR, and HR. Lab tests included blood routine (24 items), blood lipids, blood glucose (FBG, FINS, and HbA1c), hepatic and kidney function, resting blood pressure, and 12-lead resting electrocardiography. (2) Data preprocessing. All variables were combined and deduplicated based on ID. After that, the "impute" function of the R library "Hmsic" was used to handle the missing values. The variable characteristics of the study participants with missing values greater than 30% were deleted, and the variables less than 30% were replaced with the average value. (3) Data transformation. For coding categorical variables into numbers, an integer code was assigned to each category, for example, gender (which had values 0 = male, 1 = female), smoking status (which had values 0 = nonsmoker, 1 = ever smoker, 2 = present smoker), alcohol intake (which had values 0 = nondrinker, 1 = ever drinker, 2 = current drinker), psychological status (which had values 1 = no pressure, 2 = a little pressure, 3 = moderate pressure, 4 = severe pressure, 5 = extreme pressure). Body mass index (BMI) was defined as weight divided by the square of height, expressed in $kg/m^2$, and was derived from mass (in kilograms) and height (in meters). WHR was defined as the ratio of waist circumference to hip circumference. The systolic blood pressure (SBP) and diastolic blood pressure (DBP) were the averages of the three measurements.

## Statistical analysis

After data preprocessing, 63 variables (Table 1) of 2,311 participants, including 1,458 controls and 853 cases, were selected for statistical analysis. The Shapiro–Wilk W test was applied to detect whether the data came from a normal distribution. Continuous variables that came from a normal distribution were presented as the mean ± standard deviation and Student's t-test was used to compare the means for these variables in the different groups. For continuous variables not normally distributed, the median (Q1–Q3 quantiles) was displayed and analyzed with the Wilcoxon rank-sum test. For categorical variables, numbers and percentages were presented and prevalence of these variables was compared using Pearson's Chi-squared test. Due to the small sample groups in some variables, the comparison was using Fisher's exact test. Results are shown in Table 1. Next, a univariate analysis was performed as a means of identifying the predictor variables related to CHD and screened out the variables with $P < 0.05$.

## LASSO model

In this section, 2,311 participants with 63 variables (Table 1) were used to variable selection and parameter estimation. LASSO regression algorithm is useful for fitting a wide variety of models with high prediction accuracy and interpretability. LASSO shrinks the absolute sum of the coefficients (L1 regularization) to be less than a fixed value (λ) and assigns zero weights to some irrelevant features to perform variable selection and

**Table 1 Characteristics of participants according to CHD categories.**

| Variable | Non-CHD | CHD | P |
|---|---|---|---|
| Gender | | | <0.001 |
| Male | 433 (55.58) | 346 (44.42) | |
| Female | 1025 (66.91) | 507 (33.09) | |
| Age | 43.00 (32.00–51.00) | 64.00 (58.00–69.00) | <0.001 |
| Smoking | | | <0.001 |
| Never | 1201 (66.39) | 608 (33.61) | |
| Current | 220 (66.67) | 110 (33.33) | |
| Ever | 37 (21.51) | 135 (78.49) | |
| Drinking | | | <0.001 |
| Never | 1210 (64.64) | 662 (35.36) | |
| Current | 227 (67.56) | 109 (32.44) | |
| Ever | 21 (20.39) | 82 (79.61) | |
| Pressure | | | <0.001 |
| No | 847 (56.32) | 656 (43.68) | |
| Slight | 313 (77.67) | 90 (22.33) | |
| Moderate | 187 (82.38) | 40 (17.62) | |
| Severe | 89 (61.81) | 55 (38.19) | |
| Extreme | 22 (64.71) | 12 (35.29) | |
| PSQI | 3.00 (2.00–4.00) | 4.00 (2.00–6.00) | <0.001 |
| BFR | 28.10 (23.60–32.10) | 33.90 (29.00–37.90) | <0.001 |
| BMR | 1321.00 (1220.25–1462.75) | 1397.00 (1270.00–1580.00) | <0.001 |
| VFI | 6.00 (4.00–9.00) | 12.00 (9.00–15.00) | <0.001 |
| SBP | 114.33 (107.33–123.00) | 135.00 (122.00–148.33) | <0.001 |
| DBP | 73.33 (68.67–78.67) | 81.67 (75.00–89.33) | <0.001 |
| WHR | 0.83 (0.78–0.87) | 0.93 (0.87–0.98) | <0.001 |
| BMI | 42.24 (39.95–44.38) | 42.96 (40.96–45.22) | <0.001 |
| WBC ($\times 10^9$/L) | 5.40 (4.60–6.40) | 5.90 (5.10–6.90) | <0.001 |
| PLCR% | 32.70 (27.90–38.40) | 32.80 (27.30–38.40) | 0.425 |
| MONO% | 5.50 (4.60–6.50) | 5.40 (4.60–6.40) | 0.051 |
| MONO ($\times 10^9$/L) | 0.30 (0.24–0.37) | 0.32 (0.26–0.40) | <0.001 |
| HCT | 41.90 (39.20–45.40) | 43.90 (41.40–46.50) | <0.001 |
| RDW_CV | 13.10 (12.60–13.80) | 13.30 (12.80–13.80) | <0.001 |
| RBC ($\times 10^{12}$/L) | 4.59 (4.34–4.91) | 4.73 (4.44–5.04) | <0.001 |
| RDW_SD | 43.00 (41.00–45.00) | 44.00 (42.00–46.00) | <0.001 |
| LYMPH% | 33.92 ± 7.85 | 33.70 ± 8.11 | 0.506 |
| LYMPH ($\times 10^9$/L) | 1.80 (1.49–2.17) | 1.93 (1.61–2.36) | <0.001 |
| MCV | 91.50 (87.73–95.10) | 92.50 (89.60–96.50) | <0.001 |
| MCH | 29.40 (28.20–30.50) | 29.70 (28.70–30.70) | <0.001 |
| HCHC | 321.00 (311.00–329.00) | 321.00 (311.00–328.00) | 0.638 |
| MPV | 11.10 (10.50–11.70) | 11.10 (10.40–11.70) | 0.476 |
| BASO% | 0.30 (0.20–0.50) | 0.40 (0.20–0.50) | 0.019 |
| BASO ($\times 10^9$/L) | 0.022 (0.010–0.030) | 0.025 (0.010–0.030) | <0.001 |
| EOS% | 1.40 (0.80–2.20) | 1.60 (1.00–2.60) | <0.001 |

(Continued)

| Table 1 (continued) | | | |
|---|---|---|---|
| **Variable** | **Non-CHD** | **CHD** | **P** |
| EOS (×10$^9$/L) | 0.07 (0.04–0.12) | 0.10 (0.06–0.16) | <0.001 |
| HGB | 134.00 (125.00–146.00) | 140.00 (131.00–150.00) | <0.001 |
| PCT | 0.26 (0.22–0.30) | 0.25 (0.22–0.29) | <0.001 |
| PDW | 12.30 (11.30–13.80) | 12.40 (11.30–13.80) | 0.771 |
| PLT (×10$^9$/L) | 237.00 (198.25–275.75) | 228.00 (192.00–267.00) | 0.002 |
| NEUT% | 57.80 (52.50–63.90) | 58.30 (52.40–63.60) | 0.601 |
| NEUT | 3.09 (2.50–3.88) | 3.45 (2.75–4.16) | <0.001 |
| LDL_C | 2.36 (2.08–2.65) | 2.94 (2.42–3.54) | <0.001 |
| TG | 0.91 (0.71–1.17) | 1.57 (1.11–2.24) | <0.001 |
| HDL_C | 1.32 (1.18–1.52) | 1.18 (1.02–1.39) | <0.001 |
| TC | 4.40 (4.10–4.80) | 5.20 (4.50–6.00) | <0.001 |
| FBG | 5.10 (4.90–5.50) | 5.60 (5.20–6.30) | <0.001 |
| HbA1c | 5.30 (5.00–5.60) | 5.90 (5.50–6.40) | <0.001 |
| FINS | 5.80 (4.10–8.10) | 7.50 (5.10–11.20) | <0.001 |
| ALT | 15.00 (11.00–21.00) | 19.00 (15.00–26.00) | <0.001 |
| Creatinine | 58.00 (51.00–66.00) | 63.00 (54.87–73.00) | <0.001 |
| IBIL | 10.80 (8.00–14.80) | 12.10 (9.10–15.50) | <0.001 |
| ALP | 72.00 (59.00–87.00) | 87.00 (74.00–104.00) | <0.001 |
| Urea | 4.50 (3.86–5.50) | 5.10 (4.35–6.14) | <0.001 |
| Uric_acid | 257.66 (215.00–307.00) | 291.00 (243.00–346.00) | <0.001 |
| AST | 20.00 (18.00–23.00) | 22.00 (19.00–26.00) | <0.001 |
| DBIL | 3.90 (3.10–5.00) | 3.90 (3.10–5.00) | 0.721 |
| TBIL | 14.75 (11.20–19.80) | 15.90 (12.40–20.20) | <0.001 |
| THB | 14.64 (12.78–16.60) | 15.45 (13.73–17.34) | <0.001 |
| HR | 67.00 (62.00–73.00) | 67.00 (61.00–74.00) | 0.660 |
| RR | 0.885(0.815–0.965) | 0.891 (0.805–0.976) | 0.753 |
| PR | 0.148 (0.137–0.161) | 0.154 (0.142–0.168) | <0.001 |
| QRS | 0.099 (0.094–0.105) | 0.100 (0.094–0.108) | 0.006 |
| QT | 0.394 (0.379–0.411) | 0.405 (0.385–0.424) | <0.001 |
| QTc | 0.419 (0.406–0.432) | 0.429 (0.414–0.444) | <0.001 |
| SV1 + RV5 | 2.190 (1.790–2.640) | 2.390 (1.940–2.910) | <0.001 |
| ECGNOTE | | | <0.001 |
| Normal | 1305 (73.19) | 479 (26.81) | |
| Load | 102 (41.63) | 143 (58.37) | |
| Overload | 51 (18.09) | 231 (81.91) | |

**Note:**
PSQI, Pittsburgh sleep quality index; BFR, body fat rate; BMR, basal metabolic rate; VFI, visceral fat index; SBP, systolic blood pressure; DBP, diastolic blood pressure; WHR, waist-to-hip ratio; WBC, white blood cells; PLCR%, platelet ratio; MONO%, percentage of monocytes; MONO, monocyte; HCT, hematocrit; RDW-CV, variable coefficient of red cell distribution width; RBC, red blood cells; RDW-SD, a standard deviation of red cell distribution width; LYMPH%, percentage of lymphocytes; LYMPH, lymphocyte; MCV, mean corpuscular volume; MCH, mean corpuscular hemoglobin; HCHC, mean corpuscular hemoglobin concentration; MPV, mean platelet volume; BASO%, percentage of basophils; BASO, basophil; EOS%, percentage of eosinophils; EOS, eosinophil; HGB, hemoglobin; PCT, platelet hematocrit; PDW, platelet distribution width; PLT, platelet; NEUT%, percentage of neutrophils; NEUT, neutrophil; LDL_C, low-density lipoprotein cholesterol; TG, triglycerides; HDL_C, high-density lipoprotein cholesterol; TC, total cholesterol; FBG, fasting blood glucose; HbA1c, glycated hemoglobin A1c; FINS, fasting insulin levels; ALT, alanine aminotransferase; IBIL, indirect bilirubin; ALP, alkaline phosphatase; TBIL, total bilirubin; AST, aspartate aminotransferase; DBIL, direct bilirubin; THB, total hemoglobin; HR, heart rate.

parameter estimation. λ is chosen with an automated ten-fold cross-validation approach. LASSO can minimize collinearity and overfitting when multicollinearity is present among input variables (*Muthukrishnan & Rohini, 2016*; *Tibshirani, 2011*). In this research, LASSO was applied for variable selection and parameter estimation of the influencing factors for CHD. Variables with a regression coefficient of zero after shrinkage were eliminated from the model. Then the remaining variables were entered into the regression model implemented with the R package (*Friedman, Hastie & Tibshirani, 2010*).

At last, 16 predictors (BFR, WHR, TG, HbA1c, LDL_C, TC, PSQI, VFI, age, SBP, FBG, ALT, creatinine, ECGNOTE, smoking, and drinking) of 2,311 participants were screened out for developing the CHD model with the LASSO, SVM and RF.

## SVM model

SVM is a supervised learning algorithm. Its learning strategy is to seek the optimal segmentation hyperplane using support vectors and margins to classify the data. SVM can be used for classification and regression analysis. As a training algorithm, SVM has high accuracy and less prone to overfitting (*Noble, 2006*). In this research, the "tune.svm" function with an automated ten-fold cross-validation was applied for parameter tuning and the "svm" function in the "e1071" R package was used for the construction of CHD model (*Meyer et al., 2021*).

## RF model

RF is a classifier containing multiple decision trees, and its output classification result is determined by the highest votes of all the tree. It is an ensemble learning method can be used for classification and regression. RF can solve the overfitting problem that often occurs in decision trees, and get more accurate prediction models (*Breiman, 2001*; *Cutler, Cutler & Stevens, 2012*). RF, compiled in the R package "randomForest" was chosen as a classifier to predict future possibilities of CHD with GHC data (*Liaw & Wiener, 2002*). A ten-fold cross-validation procedure was employed to optimize the parameters (ntree and mtry).

## Effectiveness of risk prediction models

The performance of risk prediction models can be evaluated in various ways. In this study, the effectiveness of the three CHD models generated by LASSO, SVM, and RF classifiers were all assessed using classification accuracy, sensitivity, specificity, precision, F1-score, Matthews correlation coefficient (MCC), and AUC measures of ROC curves. Besides, the NRI and IDI were also calculated using the "reclassification" functions in the "PredictABEL" package in R (*Kundu et al., 2011*). NRI and IDI were used to assess model discrimination with the cutoff = 0.5. AUC, NRI, and IDI are three metrics that are increasingly used together in the assessment of binary outcomes models (*Martens, Tonk & Janssens, 2019*). In addition, DCA was used to evaluate the net benefit across a range of threshold probabilities among the three models developed by LASSO, SVM, and RF classifiers, and identified the optimal model (*Vickers & Elkin, 2006*).

## RESULTS

### General characteristics of the participants

After data preprocessing, 2,311 rural residents including 779 males and 1,532 females with 63 variables of Xinxiang county, Henan province in China were screened out for analysis. Results of descriptive statistics for the selected 63 variables in groups with and without CHD for the 2,311 individuals are shown in Table 1. Compared with the controls, patients with CHD were more likely to be males, older, ever-smokers, and ever-drinkers, and had higher PSQI, BFR, BMR, VFI, SBP, DBP, WHR, and BMI ($P < 0.001$). Besides, blood test results (24 items) showed that patients with CHD had a higher WBC count including five subtypes NEUT count, LYMPH count, MONO count, BASO count, EOS count, and the percentage of EOS than the control group ($P < 0.001$). The levels of RBC, HGB, RDW-CV, RDW-SD, HCT, MCV, and MCH were also elevated in patients with CHD ($P < 0.001$). In contrast, PCT and PLT counts were significantly lower in CHD patients compared to the normal group ($P < 0.001$ and $P = 0.002$). Additionally, the level of blood glucose (FBG, HbA1c, and FINS) and blood lipid (LDL_C, TG, and TC) of patients with CHD was significantly higher than that without CHD ($P < 0.001$). However, compared with the control group, the level of HDL_C was decreased in patients with CHD. All indices of hepatic and renal function (ALT, creatinine, IBIL, ALP, urea, uric_acid, AST, TBIL, and THB) were all significantly elevated in patients with CHD except DBIL. Prolonged PR and QT intervals, QRS duration, QTc, and SV1 + RV5 on resting 12 lead resting ECGs were observed in the CHD patients ($P < 0.001$). All of these variables except for RDW_CV and PR intervals that differed between the CHD group and the normal group were also statistically significant in univariate logistic regression ($P < 0.01$) (Table S1). For example, the prevalence of CHD increased with age (OR = 1.15, 95% CI = 1.14–1.16) and was higher in men than in women (OR = 0.62, 95% CI = 0.52–0.74). In addition, CHD prevalence also increased with high MONO counts (OR = 5.40, 95% CI = 2.47–11.80) and EOS counts (OR = 5.82, 95% CI = 2.72–12.43), elevated LDL_C (OR = 5.99, 95% CI = 4.98–7.20), TG (OR = 20.16, 95% CI = 15.24–26.68) and TC (OR = 5.23, 95% CI = 4.43–6.16) levels, high FBG (OR = 3.39, 95% CI = 2.91–3.95) and HbA1c (OR = 8.01, 95% CI = 6.47–9.91) levels with all $P < 0.001$.

### Parameter evaluation and feature selection

Parameters tuning for different models were performed by ten-fold cross-validation that gave the best hyper parameters with the lowest cross-validation average error. The optimal parameters combination for svm model were 0.01 for gamma and 100 for cost. The best mtry and ntree for RF model were 150 and eight. To choose the most useful explanatory variables from 63 candidate variables, first, univariate logistic regression analysis was used to preliminarily screen the variables significantly associated with CHD. A total of ten variables including PLCR%, RDW_CV, LYMPH%, HCHC, MPV, PDW, NEUT%, DBIL, HR, and RR not statistically significant associated with CHD ($P > 0.05$) were excluded (Table S1). Next, automated ten-fold cross-validation of the LASSO

**Table 2 Model performance of the three classifiers.**

| Models | Performance | Train (%) | Test (%) |
|--------|-------------|-----------|----------|
| LASSO | Accuracy | 92.12 | 89.52 |
| | Precision | 91.43 | 88.41 |
| | Sensitivity | 86.88 | 82.07 |
| | Specificity | 95.21 | 93.81 |
| | F1-score | 89.10 | 85.12 |
| | MCC | 83.00 | 77.18 |
| SVM | Accuracy | 96.61 | 91.27 |
| | Precision | 98.24 | 91.70 |
| | Sensitivity | 92.52 | 83.67 |
| | Specificity | 99.02 | 95.64 |
| | F1-score | 95.30 | 87.50 |
| | MCC | 92.75 | 81.01 |
| RF | Accuracy | 100 | 93.89 |
| | Precision | 100 | 95.63 |
| | Sensitivity | 100 | 87.25 |
| | Specificity | 100 | 97.71 |
| | F1-score | 100 | 91.25 |
| | MCC | 100 | 86.78 |

Note:
LASSO, least absolute shrinkage and selection operator regression algorithms; RF, Random Forest; SVM, Support Vector Machine.

regression (cv.glmnet) was used for selecting the most parsimonious model with the largest $\lambda$ (lambda.lse = 0.02) whose error was no more than one standard error above the error of the best model. Sixteen variables (BFR, WHR, TG, HbA1c, LDL_C, TC, PSQI, VFI, age, SBP, FBG, ALT, creatinine, ECGNOTE, smoking, and drinking) with non-zero coefficient were screened out. Among the 16 variables, WHR, TG, and ECGNOTE had the largest coefficients, namely 1.5355, 1.004, and 0.8523 respectively. In contrast, BFR, ALT, and creatinine had the smallest coefficients, which were 0.0035, 0.0031, and 0.0002 respectively. Finally, a total of 2,311 participants (1,458 controls and 853 CHD patients) including 16 variable features were selected to construct the CHD model with the LASSO, SVM, and RF methods. A training dataset (70% of the patients with CHD and 70% of the controls without CHD) was for developing the risk prediction models, and a test dataset (30% of the patients with CHD and 30% of the controls without CHD) to test the performance of the model.

## Comparison of predictive model performance

Performance assessment of the three models is presented in Table 2. Overall, the three classifiers (LASSO, SVM, and RF) performed well in categorizing CHD patients and non-CHD individuals. The three models all achieved significantly higher AUC values (>0.95). Among the three classifiers, RF attained the best performance on risk prediction of CHD, with highest accuracy (1.00 and 0.9389), sensitivity (1.00 and 0.8725), specificity

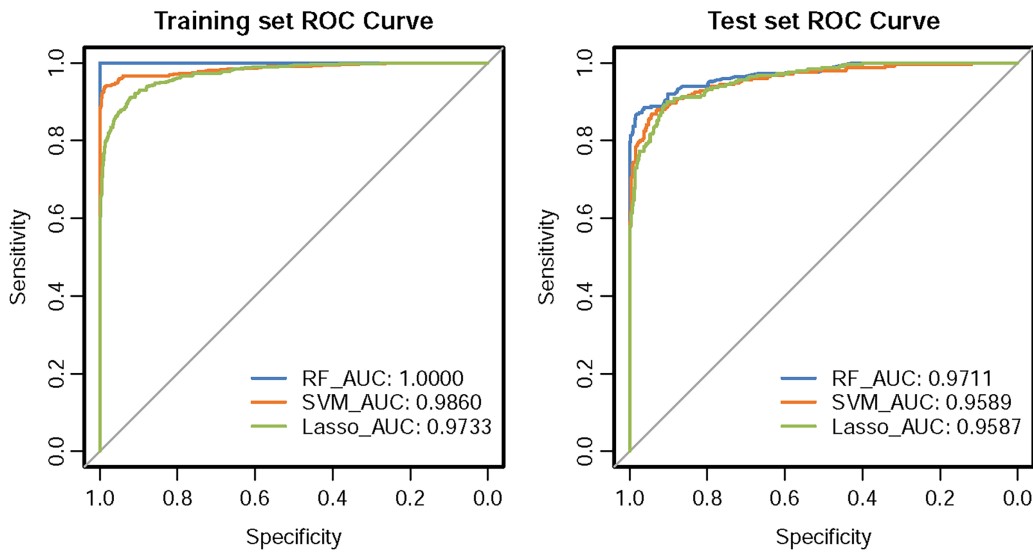

**Figure 1 ROC curves of the three classifiers.** An estimated AUC under the ROC curve of the RF, SVM and LASSO classification and prediction model for cases classified as presence of CHD and absence of CHD in the training dataset and test dataset, where the true positive rate is sensitivity on the y-axis and the true negative rate is the difference (specificity) on the x-axis.

(1.00 and 0.9771), precision (1.00 and 0.9563), F1-score (1.00 and 0.9125), and MCC (1.00 and 0.8678) on both the training set and test set, followed by SVM model, while the LASSO model performed worst. Details are shown in Table 2.

The ROC curve results of the three CHD models are shown in Fig. 1. The CHD models constructed by the RF algorithm can provide a consistently high accuracy for predicting CHD outcome, with the highest AUC both on the training set and test set (1.00 and 0.9711), followed by SVM (0.9860 and 0.9589) and LASSO (0.9733 and 0.9587). Besides, the IDI and categorical NRI were also calculated. Results showed that the RF model had significantly improvement in CHD risk prediction both on the training set (NRI = 8.52%, 95% CI = 6.32%~10.72%, $P < 0.001$ and IDI = 9.54%, 95% CI = 8.15% ~10.94%, $P < 0.001$) and the test set (NRI = 6.39%, 95% CI = 2.59%~10.19%, $P = 0.00099$ and IDI = 4.21%, 95% CI = 1.68%~6.74%, $P = 0.00112$) compared with the SVM model. The RF model was also better than the LASSO model both on the training set (NRI = 17.92%, 95% CI = 14.92%~20.92%, $P < 0.001$ and IDI = 19.50%, 95% CI = 17.71% ~21.29%, $P < 0.001$) and the test set (NRI = 8.85%, 95% CI = 5.07%~12.63%, $P < 0.001$ and IDI = 8.78%, 95% CI = 6.45%~11.11%, $P < 0.001$). The SVM model was superior to the LASSO model both on the training set (NRI = 9.40%, 95% CI = 6.88%~11.92%, $P < 0.001$ and IDI = 9.95%, 95% CI = 8.57%~11.34%, $P < 0.001$) and the test set (NRI = 2.46%, 95% CI = −1.30%~6.23%, $P = 0.19969$ and IDI = 4.57%, 95% CI = 2.58% ~6.57%, $P = 10^{-5}$), but not statistically significant of NRI on the test set. Besides, results of DCA also indicated the CHD risk prediction model developed by the RF classifier had a higher net benefit than the models constructed by SVM and Lasso methods both on the training set and test set (Fig. 2).

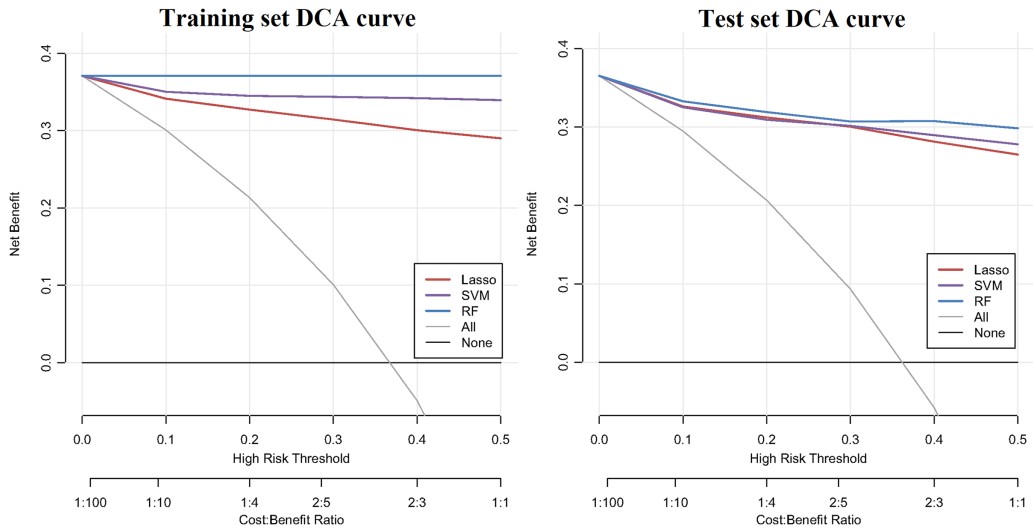

**Figure 2 DCA curve of CHD models constructed by three methods (RF, SVM and LASSO).** The y-axis shows the net benefit. The x-axis indicates the threshold probability. The grey line represents the assumption that all individuals have CHD. The solid line indicates the hypothesis that no one is diagnosed as CHD.      

# DISCUSSION

This study aims to establish a CHD prediction model suitable for a Chinese rural population from Xinxiang county, Henan province. Generally speaking, the three CHD models all performed well, and the AUC value reached more than 95%. Among the three models, RF performed best with the highest values in ten performance metrics (accuracy, precision, sensitivity, specificity, F1-score, MCC, AUC, NRI, IDI and DCA) on the training set as well as the test set, followed by SVM, and the LASSO model performed worst.

Early diagnosis and intervention of CHD plays an important role in reducing the death caused by this disease (*Adams, Bojara & Schunk, 2018*). There are many ways to diagnose CHD, including exercise electrocardiogram (*Singh et al., 2020*), echocardiography (*Mertes et al., 1991*), coronary angiography (*Newby et al., 2015*; *Williams et al., 2016*), and so on. Among them, coronary angiography can directly observe the location of coronary artery lesions, the degree of stenosis and the smoothness of distal blood flow. It is considered to be the gold standard for the diagnosis of CHD (*Wu et al., 2019*). However, conventional coronary angiography is an invasive examination method, and the cost of computed tomography coronary angiography is relatively high (*Darlington et al., 2015*). In rural areas of China, due to the limited medical level, this CHD diagnosis method has not been widely used. Although exercise ECG is a non-invasive and low-cost diagnostic method for CHD, its low sensitivity can easily lead to misdiagnosis (*Singh et al., 2020*). Therefore, it is necessary to develop a highly sensitive, accurate and low-cost diagnostic method for CHD for rural residents in Henan, China. This study combined the easily accessible health examination indicators and questionnaire survey data of residents in Xinxiang County, Henan Province, and used machine learning

methods to construct a CHD risk prediction model. Among them, the RF model performed best, with the highest accuracy and AUC (93.89% and 0.9711), followed by the SVM model (91.27% and 0.9589), and LASSO model (89.52% and 0.9587). It demonstrated that RF classifier applied to routine physical examination data could be used to predict the risk of CHD and provide assistance to the local disease control center in formulating preventive measures for CHD.

So far, numerous CHD risk prediction models have been established. Framingham risk scales, ACC/AHA, QRISK, QRISK2, and SCORE were constructed from population cohort data and widely used for disease prevention (*Collins & Altman, 2012*; *Goff et al., 2013*; *Wilson, D'Agostino & Levy, 1998*). However, due to the new risk factors of CHD discovered and machine learning applications in healthcare, various new CHD risk models using machine learning algorithms were also constructed. For example, *Kathleen, Julia & George (2016)* developed four CHD diagnosis and outcome prediction models using an adaptive boosting algorithm with four different data sets from Cleveland Clinic Foundation (CCF), Hungarian Institute of Cardiology (HIC), Long Beach Medical Center (LBMC), and Switzerland University Hospital (SUH). Results showed that AUCs and accuracies for CCF, HIC, LBMC, and SUH CHD models utilizing 28 input attributes were 0.8526 and 80.14%, 0.9212 and 89.12%, 0.6864 and 77.78%, and 0.6357 and 96.72%, respectively (*Kathleen, Julia & George, 2016*). In another Korean study, Jae et al developed a neural network-based CHD risk prediction model using nine features with AUC 0.749 and 82.51% (*Kim & Kang, 2017*). In addition, *Beunza et al. (2019)* constructed CHD risk prediction models for 4,240 participants from the open database of the Framingham Heart Study using five different algorithms, decision tree, RF, SVM, neural networks, and logistic regression implemented with two different software tools. In their study, they found that neural networks attained the best performance in the prediction of CHD incidents with AUC = 0.71 when analyzing the data in R-Studio, and the SVM performed best with AUC = 0.75 when analyzing the data in RapidMiner. In this study, 2,311 rural residents with 16 variables of Xinxiang county, Henan province in China were screened out for developing the CHD risk prediction models using two machine learning algorithms (SVM and RF), with LASSO as a baseline comparison. The models constructed by the three classifiers (RF, SVM, and LASSO) performed well in the prediction of CHD risk, with accuracy and AUC values greater than 89% and 0.95 on both the training set and the test set. Compared to previous CHD models, fewer features were utilized and better prediction performances were obtained of our models.

According to the China Health Statistics Yearbook in 2018, the death rate of CHD for urban residents was 115.32 per 100,000, and the death rate of CHD for rural residents was 122.04 per 100,000 in 2017. The mortality rate of CHD has been increasing since 2012, and the mortality rate in rural areas has increased significantly, which has exceeded the urban level by 2016 (*China NBoSo, 2018*; *Ma et al., 2020*). In the United States, CHD affected 18,200,000 people (age ≥ 20), with a prevalence of 6.7%, and there were 365,914 deaths from CHD in 2017. From 2006 to 2016, the annual mortality rate of CHD among the American population declined 31.8%. In addition, the mortality rate of CHD in urban areas was lower than that in rural areas (*Benjamin et al., 2019*). There are many

reasons for the difference in CHD incidence and mortality between China and the United States, such as geographical location, eating habits, ethnic differences, economic level, and so on (*Bhatnagar, 2017*; *Kreatsoulas & Anand, 2010*; *Mirzaei et al., 2009*). Here we mainly discuss the impact of CHD risk factors on this difference. Smoking, high blood pressure, high levels of TC, LDL-C, HDL-C, diabetes mellitus, and age were well-established independent risk factors of CHD, which had been elucidated by the Framingham Heart Study in US (*Wilson, D'Agostino & Levy, 1998*). Besides, obesity and lack of exercise were also considered as risk factors for CHD by the AHA (*Bogers et al., 2007*; *Powell-Wiley et al., 2021*). The decline in mortality from CHD in the United States is largely attributable to changes in CHD risk factors, including reductions in blood cholesterol, SBP, smoking prevalence, physical inactivity, and improvements in diagnosis and treatment (*Kulshreshtha et al., 2014*; *Singh et al., 2019*). Compared with populations in US, the increase of CHD death rate in China is attributable to lifestyle and living conditions changes caused by economic development, from carbohydrates for energy and heavy labouring work to protein, lipid for energy and sedentary with economic development. Hypertension, smoking, overweight, hypercholesterolaemia (total cholesterol > 5.20 mmol/l), and diabetes became more prevalent and promoted the incidence of CHD in the Chinese population. Although the five risk factors for CHD in the Chinese population were consistent with those in the US population, the total serum cholesterol level in the Chinese population was lower than that in the US population (*Lu et al., 2018*; *Zhang, Lu & Liu, 2008*). In addition, in both the United States and China, mortality from CHD was higher in rural areas than in urban areas, which was closely related to socioeconomic status (*O'Connor & Wellenius, 2012*). In this research, sixteen variables (BFR, WHR, TG, HbA1c, LDL_C, TC, PSQI, VFI, age, SBP, FBG, ALT, creatinine, ECGNOTE, smoking, and drinking) with nonzero coefficients were considered as the predictors for CHD of rural residents of Xinxiang County, Henan Province. Among the 16 predictors, BFR, WHR, FBG, TG, LDL_C, TC, VFI, age, SBP, ALT, creatinine and smoking have been confirmed to be related to the onset of CHD (*Emdin et al., 2017*; *Klempfner et al., 2016*). However, a study conducted in Iran showed the use of ECG to predict coronary artery stenosis was poor, and the sensitivity and specificity were 51.5% and 66.1%, respectively (*Mahmoodzadeh et al., 2011*). Besides, a study conducted on Singaporean non-diabetic Chinese indicated that there was a significant association between creatinine and CHD, but no significant association was seen between HbA1c and CHD (*Salim et al., 2020*). Moreover, *Tolstrup et al. (2006)* observed a negative association between alcohol intake and CHD for women, whereas among men, drinking frequency inverse associated with the risk of CHD in Denmark. Whether these factors are risk factors for CHD needs further study.

The data of this study came from a cross-sectional survey of common chronic noncommunicable diseases in rural areas of Henan Province in China. There was a lot of missing, duplicate, and unrelated information in the datasets. To achieve the best performance, data pre-processing was performed. Among the variables in our datasets,

there were couples of variables highly correlated, such as IBIL and TBIL, ALT and AST, SBP and DBP, HCT and RBC, HCT and HGB, RBC and HGB, LDL_C and TC, MCV and MCH. Therefore, LASSO was used for features selection. LASSO performs well on the variable selection of the datasets with multicollinearity and high dimensionality among the variables. The LASSO method uses a regularization method to constrain the parameters of the model. It sets an upper limit for the sum of the absolute values of all parameters and shrinks some parameters in the model to zero. At last, features with non-zero coefficients in the model are retained. This process can filter features and reduce prediction errors (*Muthukrishnan & Rohini, 2016*; *Tibshirani, 2011*).

To reduce the burden of CHD in Xinxiang County, Henan Province, on the one hand, government agencies need to improve the level and quality of medical care and strengthen the control of CHD risk factors. On the other hand, for those high-risk groups who have smoking or drinking habits, high waist-to-hip ratio, poor sleep quality, high creatinine, high blood sugar, hyperlipidemia and the elderly, it is necessary to vigorously promote health knowledge and actively control behavioral risk factors, such as avoiding unhealthy diet, engaging in regular physical activity, etc. In addition, local physician need to further confirm the CHD patients predicted by the model in this study, especially those with hypertension, hyperglycemia, hyperlipidemia, overweight and smoking history, and make sure they have access to early diagnosis and treatment.

Some limitations of this study included the following: First, although the overall sample size in this research was large, there was a small number of CHD cases, and the participants were mainly from two communities in Xinxiang County, Henan Province. More participants from different regions and races are needed. Second, the CHD models in this study were developed with a cross-sectional dataset. Accurate prediction of CHD risk depends on long-term prospective cohort studies. Third, outcome events were obtained through a questionnaire survey, which may have some influence on the result. In future studies, we will complete the collection of this cohort data and verify these results.

## CONCLUSIONS

In conclusion, we developed a CHD risk prediction model for Xinxiang county rural residents and compared the estimates from the model with the other CHD models. With limited medical conditions in rural areas, physical examinations are cheaper and more accessible than coronary angiography. Therefore, in this study, physical examination indicators combined with machine learning method were used to construct a CHD risk prediction model. Results showed that the CHD model constructed in this study performed well and had higher accuracy, precision and AUC compared with the previous models. Thus using machine learning methods combined with health examination indicators to build models for complex disease diagnosis is promising. In addition, with the gradual rise in CHD events, the prevention and treatment of CHD influencing factors such as BFR, WHR, TG, HbA1c, LDL_C, TC, PSQI, VFI, age, SBP, FBG, ALT, creatinine, ECGNOTE, smoking, and drinking are important public health concerns.

### Funding

This work was supported by the National Key Research and Development Program of China (2016YFC0900803), the National Natural Science Foundation of China (Grant No. 82001117), and the Scientific and Technological Development Foundation of Henan province (China) (Grant No.202102310401). The funders had no role in study design, data collection and analysis, decision to publish, or preparation of the manuscript.

### Grant Disclosures

The following grant information was disclosed by the authors:
National Key Research and Development Program of China: 2016YFC0900803.
National Natural Science Foundation of China: 82001117.
Scientific and Technological Development Foundation: 202102310401.

### Competing Interests

The authors declare that they have no competing interests.

### Author Contributions

- Qian Wang conceived and designed the experiments, performed the experiments, analyzed the data, prepared figures and/or tables, authored or reviewed drafts of the paper, and approved the final draft.
- Wenxing Li conceived and designed the experiments, performed the experiments, analyzed the data, prepared figures and/or tables, authored or reviewed drafts of the paper, and approved the final draft.
- Yongbin Wang performed the experiments, analyzed the data, prepared figures and/or tables, and approved the final draft.
- Huijun Li performed the experiments, prepared figures and/or tables, and approved the final draft.
- Desheng Zhai performed the experiments, analyzed the data, prepared figures and/or tables, and approved the final draft.
- Weidong Wu conceived and designed the experiments, analyzed the data, authored or reviewed drafts of the paper, and approved the final draft.

### Human Ethics

The following information was supplied relating to ethical approvals (*i.e.*, approving body and any reference numbers):

Ethical approval was obtained from the Xinxiang Medical University Life Science Ethics Committee (XYLL-2016176).

### Data Availability

The raw data and code are available in the Supplemental Files.

## Supplemental Information

Supplemental information for this article can be found online at http://dx.doi.org/10.7717/peerj.12259#supplemental-information.

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
