# Peer review of "Prediction of coronary heart disease in rural Chinese adults: a cross sectional study"

_PeerJ, doi:10.7717/peerj.12259_

## Round 0.1 · original submission · Major Revisions

I agree with the points raised by two reviewers. Each comment should be addressed appropriately.

·

Basic reporting

it is a very well written and focused article on a ver unique and necessary topic to address human health and urban design. the manuscript needs to update the references list on recent studies

Experimental design

very clear and well organised , just reproducibilities of data and resolution needs more elaboration

Validity of the findings

No comments

Additional comments

The paper carries interesting and important topic that is neglected in modern urbanisation. I suggest authors update their reference list based on recent studies and present the data based on each method separately to have a clear discussion

Reviewer 2 ·

Basic reporting

The study is a welcome addition to the clinical cardiovascular literature in to address an important notion of rural geographical areas vs urban areas as published in the US, UK etc.

The Chinese researchers have done excellent job from the ML technical demonstration, etc. However I feel that is a jewel hidden under ground due to the lack of clinical interpretation of this work from a practicing cardiologist standpoint.

For example, I would like to know more clearly how the CHD prediction differs in rural areas from those in urban areas. What are the important parameters or model features which would help the practicing cardiologist or provider in treating their. You are leaving me to guess this out. I would like to see this clearly stated and debated.

I feel that the discussion section is very weak with the above in mind. Remember that you are building a predictive mode in a clinical area so that I can use it in my practice.

The paper in its current format is not really providing the clarity which would allow to benefit scientifically from this rich data set.

Experimental design

Everything looks fine. Perhaps your paper may benefit from decision curve analysis so that I can get a pure estimate of true positives after discounting for the false positive.

Validity of the findings

I am not able to assess the findings in the current paper format due to the non rigor of discussion of current findings in the rural geographical areas relative to urban areas etc.

Additional comments

This is a good piece of work. But I feel that not enough work has been into this work to extract the clinical insights from a cardiologist or provider standpoint. A more elaborate treatment of the subject is highly recommended to bring clarity to the manuscript so that the reader would benefit from this rich data set.

---

## Round 0.2 · Minor Revisions

I agree that the authors responded to the comments raised by the reviewers. However, they pointed out further concerns to be addressed then I believe another round would be required.

·

Basic reporting

The work improved but the discussion section has lack of citations and critical thinking

Experimental design

well established

Validity of the findings

the impact of work can bring new knowledge to healthcare practitioners however the conclusion is too brief and needs to present how the study objectives were obtained and important to the nature of work

Additional comments

In General the work quality improved but I am concerned about discussion section

Reviewer 2 ·

Basic reporting

Comments to Editor/Authors:

The authors have made a serious attempt to improve the manuscript. However I would like to see the following changes to be made prior to final publication:

1. In my prior comment, I meant to compare the findings from the Chinese rural areas to the Western Urban areas. Please elaborate on this point as you are in a very good position to make a significant contribution about informing us about the risk factors leading to the differences in CHD morbidity between the Chinese and Western countries (e.g., UK/US).
2. You have talked about preventative efforts for CHD, can you please consider discussing the impactful variables from your model features which allow the treating physician to achieve positive intervention effects with the patient.
3. Please add a statement about the clinical insights of the study in the abstract.
4. This comment is outside my expertise. The authors are reporting an ROC =1 for training data. I do not believe that this is possible. You cannot build a perfect model even from a cross sectional cohort. There must be something wrong. Please investigate.

Experimental design

Nothing to report

Validity of the findings

Comments to Editor/Authors:

The authors have made a serious attempt to improve the manuscript. However I would like to see the following changes to be made prior to final publication:

1. In my prior comment, I meant to compare the findings from the Chinese rural areas to the Western Urban areas. Please elaborate on this point as you are in a very good position to make a significant contribution about informing us about the risk factors leading to the differences in CHD morbidity between the Chinese and Western countries (e.g., UK/US).
2. You have talked about preventative efforts for CHD, can you please consider discussing the impactful variables from your model features which allow the treating physician to achieve positive intervention effects with the patient.
3. Please add a statement about the clinical insights of the study in the abstract.
4. This comment is outside my expertise. The authors are reporting an ROC =1 for training data. I do not believe that this is possible. You cannot build a perfect model even from a cross sectional cohort. There must be something wrong. Please investigate.

---

## Round 0.3 · accepted · Accept

Regretfully both reviewers are now unavailable, however, I think that the authors responded to all the comments from them appropriately and the manuscript is ready for publication now.